# Traumatic and Diabetic Schwann Cell Demyelination Is Triggered by a Transient Mitochondrial Calcium Release through Voltage Dependent Anion Channel 1

**DOI:** 10.3390/biomedicines10061447

**Published:** 2022-06-19

**Authors:** Nicolas Tricaud, Benoit Gautier, Jade Berthelot, Sergio Gonzalez, Gerben Van Hameren

**Affiliations:** 1Institut des Neurosciences de Montpellier, Univ. Montpellier, INSERM, 34000 Montpellier, France; benoit.gautier@inserm.fr (B.G.); jade.berthelot@enscm.fr (J.B.); sergiogonzalez@invivex.com (S.G.); gvanhameren@dal.ca (G.V.H.); 2I-Stem, UEVE/UPS U861, INSERM U861, AFM, 91100 Corbeil-Essonnes, France

**Keywords:** myelin, peripheral nerve, mitochondria, VDAC

## Abstract

A large number of peripheral neuropathies, among which are traumatic and diabetic peripheral neuropathies, result from the degeneration of the myelin sheath, a process called demyelination. Demyelination does not result from Schwann cell death but from Schwann cell dedifferentiation, which includes reprograming and several catabolic and anabolic events. Starting around 4 h after nerve injury, activation of MAPK/cJun pathways is the earliest characterized step of this dedifferentiation program. Here we show, using real-time in vivo imaging, that Schwann cell mitochondrial pH, motility and calcium content are altered as soon as one hour after nerve injury. Mitochondrial calcium release occurred through the VDAC outer membrane channel and mPTP inner membrane channel. This calcium influx in the cytoplasm induced Schwann-cell demyelination via MAPK/c-Jun activation. Blocking calcium release through VDAC silencing or VDAC inhibitor TRO19622 prevented demyelination. We found that the kinetics of mitochondrial calcium release upon nerve injury were altered in the Schwann cells of diabetic mice suggesting a permanent leak of mitochondrial calcium in the cytoplasm. TRO19622 treatment alleviated peripheral nerve defects and motor deficit in diabetic mice. Together, these data indicate that mitochondrial calcium homeostasis is instrumental in the Schwann cell demyelination program and that blocking VDAC constitutes a molecular basis for developing anti-demyelinating drugs for diabetic peripheral neuropathy.

## 1. Introduction

Myelinating Schwann cells (mSC) are responsible for myelin production in the peripheral nervous system (PNS). These cells individually wrap axons with many layers of compacted and modified plasma membrane in order to form a myelin sheath that allows for the fast saltatory propagation of action potentials and axonal metabolic support [1,2]. 

Even when a complete myelin sheath has been formed around peripheral nerve axons, it remains plastic. Indeed, following peripheral nerve injury and axonal degeneration, mSC terminate their myelin sheath, entering a dedifferentiation program called “demyelination” [3]. The myelin is digested by mSC themselves or by macrophages that are recruited to the wound [4]. When axons have grown back, a re-myelination process occurs through dedifferentiated Schwann cells [5]. Besides nerve injury, mSC demyelination also occurs in hereditary and acquired demyelinating diseases [6]. Acquired demyelinating diseases include diabetic peripheral neuropathy [7], drug-related peripheral neuropathies, leprosy and peripheral neuropathies of inflammatory etiology [8]. Hereditary demyelinating diseases of the PNS, also called Charcot–Marie–Tooth diseases, are rare, but remain among the most common hereditary diseases [9]. While they are rarely lethal, these diseases range from life threatening to severely disabling and therefore put a high burden on public health systems. Thus, an important challenge is to understand the cellular and molecular events that underlie the transition from myelinating to demyelinating in Schwann cells. 

Traumatic demyelination constitutes a relevant model for studying mSC demyelination programs. After nerve injury, one of the first events to occur in mSC is the activation of mitogen-activated protein kinases (MAPK) pathways, extracellular signal-related kinases (ERK), c-Jun Nterminal kinases (JNK) and p38 mitogen-activated kinases (P38) [3], followed by cJun phosphorylation and relocation in the nucleus, triggering reprograming and myelin degeneration [10]. However, what triggers these molecular events remains unclear. We hypothesized that mitochondria participate in this signaling.

Indeed, several Schwann cell-specific KO mouse models have shown that mitochondrial dysfunctions result in peripheral myelin defects and demyelinating peripheral neuropathies: deletion of mitochondrial protein prohibitin 1 *Phb1* [11,12], mitochondrial transcription factor *Tfam* [13], the respiratory chain component *Cox10* [14], the nicotinamide mononucleotide synthetizing enzyme *Nampt* [15], or the nutrient-sensing O-linked N-acetylglucosamine transferase *Ogt* [16]. Moreover, several studies showed that mitochondrial dysfunctions are involved in an increasing number of demyelinating neurodegenerative diseases [14,17]. Patients as well as mouse models suffering from mitochondrial multisystem disorders, such as dominant optic atrophy, often show a peripheral neuropathy in addition to other debilitating symptoms [18,19,20]. The nerves of mice mimicking dominant optic atrophy display features of demyelination, altered myelin sheath as well as abnormal mitochondria [19,20]. Some acquired PNS diseases also appear to be linked to defects in mitochondrial functions. Indeed, a side effect of some anti-HIV drugs is demyelinating peripheral neuropathy [21]. In addition, this drug-induced pathology and diabetic peripheral neuropathy have been linked with perturbation of mitochondrial functions and to mitochondrial stress and defects in PNS [22]. Taken together, this suggests that mitochondria can play an essential role in demyelinating diseases of peripheral nerves. However, knowledge of how mitochondria and their morphological and physiological properties change during demyelination remains elusive.

The goal of this project was to investigate the role of mSC mitochondria during traumatic and diabetic peripheral nerve demyelination. Using in vivo fluorescent probes and multiphoton live imaging, we show here that acutely after triggering demyelination, mSC mitochondria release their calcium in the cytoplasm through voltage-dependent anion channel 1 (VDAC1) and the mitochondrial permeability transition pore (mPTP) channels, which leads to the activation of the demyelination program. In diabetic *db/db* mice, a similar transitory release of mitochondrial calcium occurs following injury, but the kinetics of this release are significantly modified, suggesting that VDAC1 channels are already leaky in the diabetic condition prior to injury. This correlates with an increase of cytoplasmic calcium in mSC of these mice. Reducing VDAC1-mediated calcium release through silencing or TRO19622, an inhibitor of this channel, prevented demyelination in traumatic conditions and reduced both peripheral nerve defects and the motor deficit in *db/db* diabetic mice. This suggests that mitochondrial calcium release through VDAC1 is intrinsically high in *db/db* mSC, and that reducing this leak may prevent diabetes-induced demyelination. 

## 2. Materials and Methods

### 2.1. Animal Housing

Immunodeficient strain CB17/SCID mice (Janvier Labs, Le Genest-Saint-Isle, France) and *db/db* diabetic mouse strain (BKS.Cg-*Dock7^m^*+/+*Lepr^db^* J Genetically Engineered Inbred, The Jackson Laboratory) were kept in the animal house facility of the Institute for Neurosciences of Montpellier in ventilated and clear plastic boxes, subjected to standard light cycles (12 h to 90 lux light, 12 h dark). *db/db* mice are used to model phase 1 to 3 of diabetes type II and obesity. Mice homozygous for the diabetes spontaneous mutation (*Lepr^db^*) demonstrate morbid obesity, chronic hyperglycemia, pancreatic beta cell atrophy and hypoinsulinemic. The care, breeding and use of animals followed the animal welfare guidelines of the “Institut National de la Santé et de la Recherche Medicale” (INSERM), under the approval of the French “Ministère de l’Alimentation, de l’Agriculture et de la Pêche”, Approval Number CEEA-LR-11032). 

### 2.2. Cloning

Plasmid dsRed2-mito7 (Clontech #55838, Mountain View, CA, USA) was cut using NheI and NotI enzymes and then treated with DNA polymerase I Large Klenow fragment to isolate mito-dsRed2 cDNA. After purification, it was cloned into pAdtrack-CMV (Quantum Biotechnologies, Inc., Montreal, QC, Canada) or pAAV-MCS (Cell Biolabs, Inc., San Diego, CA, USA) plasmids under the control of a CMV or a CAG promoter, respectively. pcDNA3.1 mito-GCaMP2 (kindly provided by Dr. X. Wang, Peking University, China) was cut or with HindIII and EcoRV to be cloned into a pShuttle-CMV (Quantum Biotechnologies, Inc.) or with BamHI and EcoRV to be cloned into a pAAV-MCS vector under the control of a CMV or a CAG promoter, respectively. Then, the GCaMP2 probe cDNA (without mito sequence) was cut using HindIII and EcoRV to be cloned into a pAAV-MCS or a pShuttle-CMV plasmid. The probe mito-SypHer (kindly provided by Dr. J.C. Jonas, Université Catholique de Louvain, Belgium) was cut using NheI and NotI enzymes and treated with DNA polymerase I Large Klenow fragment. After purification, it was cloned into pAAV-MCS under the control of a CAG promoter. The mouse VDAC1-specific shRNA sequence 2 GTTGGCTATAAGACGGATGAACT (Sigma-Aldrich, Ref. #TRCN0000012391, Saint-Quentin-Fallavier Cedex, France)**,** the VDAC1 shRNA sequence 3 ACCAGGTATCAAACTGACGTTCT (Sigma-Aldrich, Ref. #TRCN0000012392) or the shRNA control (dsRed2) AGTTCCAGTACGGCTCCAA or (GFP) CAAGCTGACCCTGAAGTTC were first cloned separately into a pSicoR vector (Addgene, Ref. 11579) under the control of a U6 promoter using HpaI and BstEII enzymes. Then, the U6-VDAC1-shRNA sequences were cut using ApaI and BstEI to be cloned into a pAAV-CMV-GFP vector (Cell Biolabs, Inc.), the pAAV-mito-GCaMP2, the pAAV-GCaMP2, the pAAV-mito-dsRed2 or the pAAV-mito-SypHer previously described. All clones were validated by sequencing. 

### 2.3. Viral Particles Production

Adenoviral particles production was described in He et al., 1998 [23]. Briefly, for adenovirus production, pAdtrack vector containing the constructs was recombined with pAdeasy1 vector in the Adeasy1 BJ5183 bacteria strain (Stratagene, La Jolla, CA, USA). The isolated adenoviral DNA was cut with PacI enzyme and transfected in HEK 293 cells using Lipofectamine 2000. The initial production of adenovirus was followed by 3 rounds of amplification. Finally, freeze–thaw cycles were used to harvest the viral particles from cells, and then they were purified using cesium chloride gradients. To produce high-titer adeno-associated virus (AAV10), three 15 cm dishes of 70–80% confluent HEK293T cells were transfected with 71 μg of pAAV expression vector, 20 μg of pAAV10 capsid and 40 μg of pHelper (Cell Biolabs, Inc., San Diego, USA). Following 48 h after transfection, the medium was collected, pooled and centrifuged for 15 min at 2000 rpm to spin down floating cells. In parallel, cells were scraped and collected in PBS. Then, cells were lysed using dry ice/ethanol bath and centrifuged for 15 min at 5000 rpm to discard cell debris. The cleared supernatant and the cleared medium were pooled and filtrated using a 0.22 μm filter. The viral solution was filtrated through a cation-exchange membrane Mustang S acrodisc (Pall Corporation, Saint-Germain-en-Laye, France) to deplete empty particles and was later filtrated through an anion-exchange membrane Mustang Q acrodisc (Pall Corporation, Saint-Germain-en-Laye, France to retain AAV viral particles. Next, viruses were eluted and concentrated using centrifugal concentrators Amicon tube. Usual titer is around 10^11^ PFU/mL. For further details, see Okada et al., 2009 [24].

### 2.4. In Vivo Virus Injection in the Sciatic Nerve

Five- to seven-week-old mice were anesthetized with isoflurane inhalation and placed under a Stemi2000 microscope (Carl Zeiss Microscopy, Rueil Malmaison, France). The incision area was shaved and cleaned using betadine solution. After incision, the *gluteus superficialis* and *biceps femoris* muscles were separated to reveal the cavity traversed by the sciatic nerve. The nerve was lifted out using a spatula, and a thin glass needle filled with viral solution (8 μL) was introduced into the nerve with a micromanipulator. This solution was injected over 30 min with short pressure pulses using a Picospritzer III (Parker Hannifin, Contamine-sur-Arve, France) coupled to a pulse generator. After injection, the nerve was replaced into the cavity, the muscles were readjusted, and the wound was closed using clips (for further details, see Gonzalez et al., 2014 [25]).

### 2.5. Sciatic Nerve Set Up under Multiphoton Microscope

Eight- to eleven-week-old injected mice were anesthetized with 5% isoflurane and 1.5% oxygen into the anesthesia system box (EZ-B800, World Precision Instruments, Hitchin, UK) for 5 min. Then, anesthesia was maintained using an anesthesia mask and the incision area was shaved and cleaned with betadine and ethanol 70% solution. Incision was realized using a scalpel, and the skin was retracted using forceps in order to expose the *gluteus superficialis* and *bíceps femoris* muscles. Next, the connective tissue that connects both muscles was cut and the sciatic nerve was gently lifted out using a spatula. A flexible bridge was slid below the sciatic nerve, and it was placed into a plastic first pool fixed to the bridge and filled with PBS. Mice were placed under the multiphoton microscope, the bridge was fixed using magnetic brackets to avoid physiological movement, and mouse legs were fixed using clippers. The microscope dark box temperature was controlled to 37 °C during all time-lapse imaging. Finally, a second pool was fixed to the first pool using a drop of agarose low melting 3% (V2111, Promega, Charbonnieres les bains, France) in Leibovitz’s L15 medium (Gibco Life Technologies) and filled with deionized water to immerse the objectives ×20 or ×63 (Carl Zeiss Microscopy, Rueil Malmaison, France). Sciatic nerves were crushed using serrated forceps at least 5 mm above the imaging area. Five successive crushes were performed at different angles to maximize the demyelination all around the nerve. 

### 2.6. Multiphoton Image Acquisition and Analysis

All time-lapse images were obtained with a multiphoton microscope Zeiss LSM 7 MP OPO. Mitochondria motility images were acquired by time-lapse recording of one image every five minutes during five hours using mito-Dsred2-labelled mitochondria (excitation light wavelength 920 nm). GCaMP2 and SypHer probe images were acquired by time-lapse recording of one image every fifteen minutes over five hours (excitation light wavelength 985 nm). Images were acquired with constant laser intensity (1%), 100 ms of acquisition time, 512 × 512 resolution and 10 images per z-stack (20 µm). Images were saved in.czi format and processed using the Image J program. Analysis was done as described in Gonzalez et al., 2015 [26].

### 2.7. Immunohistochemistry

The right sciatic nerves of eight-week-old mice were crushed as described previously. Four to twelve hours later, these nerves were dissected and washed in L15 medium, fixed in Zamboni’s fixative for 10 min at room temperature, washed in PBS, and incubated in successive glycerol baths (15, 45, 60, 66% in PBS) for 18 to 24 h each, before freezing at −20 °C. The nerves were cut in small pieces in 66% glycerol and the perineurium sheath removed. Small bundles of fibers were teased in double-distilled water on Superfrost slides, dried overnight at room temperature, and the slides stored at −20 °C. For immunostaining, the teased fibers were incubated for 1 h at room temperature in blocking solution (10% goat serum, 0.2% TritonX100, and 0.01% sodium azide in PBS). Then, the samples were incubated with anti- ECCD2 primary mouse antibody (1/100, BD biosciences, Ref. 610181, Le Pont de Claix Cedex, France), anti-phosphoS63-c-jun primary mouse antibody (1/200, BD Biosciences, Ref. 558036, Le Pont de Claix Cedex, France), anti-c-jun primary mouse antibody (1/200, BD Biosciences, Ref. 610326, Le Pont de Claix Cedex, France), anti-VDAC primary rabbit antibody (1/100, Cell Signaling, Ref. 4866, Ozyme, France) or/and MitoTracker Red (1/1000, Molecular Probes, Ref. M7515, Thermo Fischer Scientific, Nimes, France) in blocking solution overnight at 4 °C. The next day, the samples were washed in PBS and incubated for 1 h at room temperature with secondary donkey antibodies coupled to Alexa488, Alexa594 or Alexa647 (1/1600, Molecular probes, Thermo Fischer Scientific, Nimes, France) and TOPRO3 iodide (50 μM, Invitrogen, Ref. T3605, Thermo Fischer Scientific, Nimes, France). Finally, the samples were washed in PBS and mounted in Immu-mount (Thermo Fischer Scientific, Nimes, France). Images were acquired at room temperature using a 20× or 40× objective, a Zeiss confocal microscope LSM710, and its associated software. 

### 2.8. Intranerve Drug Administration

TRO19622 drug (Tocris bioscience, Ref. 2906, Bio-Techne SAS, Noyal Châtillon sur Seiche, France) was diluted in ethanol to 20 mM and then diluted in sterile PBS to 20 µM. Intrasciatic TRO19622 treatment for live imaging experiments was realized by intra-sciatic nerve injection of 2 μL of the 20 µM drug solution using a Hamilton syringe or a glass needle held on a micromanipulator as described for the virus injection. This injection occurred 30 min before multiphoton imaging or nerve injury followed by multiphoton imaging. Intraperitoneal TRO19622 treatment was realized using the 20 mM solution at 3 mg/kg. Methyl jasmonate (Sigma-Aldrich, Ref. 392707, Saint-Quentin-Fallavier Cedex, France) was diluted in 1 mL of sterile PBS to 57 µM. Methyl jasmonate treatment was realized by injection of 7 µL of MJ commercial solution (30 µmol) into the sciatic nerve using a glass needle held on a micromanipulator two hours before multiphoton image acquisition or 4 days before CARS imaging. Auranofin (Sigma Aldrich, Ref. A6733, Saint-Quentin-Fallavier Cedex, France) was diluted in DMSO to 10 mM and then diluted in sterile PBS to 2 μM. Auranofin treatment was realized by intra-sciatic nerve injection of 2 μL of solution using a Hamilton syringe during multiphoton image acquisition. Cyclosporine A (Sigma Aldrich, Ref. 30024-25MG, Saint-Quentin-Fallavier Cedex, France) was diluted to 100 mM in ethanol then diluted in sterile PBS to 50 μM and 500 μM. Cyclosporine A treatment was realized by intra-sciatic nerve injection of 4 μL of solution using a glass needle and micromanipulator 30 min before multiphoton image acquisitions. Ethanol for TRO19622 or PBS for methyl jasmonate and Cyclosporine A were used as vehicle. 

### 2.9. Systemic Drug Administration 

TRO19622 (Bio-techne, Ref. 2906, Noyal Châtillon sur Seiche, France) was diluted to 0,3 mg/mL in Cremophore EL/dimethylsulfoxide/ethanol/phosphate buffer saline (CDEP, 5/5/10/80 *v*/*v*). TRO19622 treatment (3 mg/kg) was performed daily by subcutaneous injection. Control mice were injected the same way with CDEP (5/5/10/80, *v*/*v*). 

### 2.10. CARS Imaging and Image Analysis

Four days after MJ injection or nerve crush, animals were sacrificed, and their sciatic nerves were collected, washed by PBS and fixed for 1 h in 4% PFA solution at room temperature. All CARS images were obtained as described previously [27] with a two-photon microscope LSM 7 MP coupled to an OPO (Carl Zeiss Microscopy, Rueil Malmaison, France) and complemented by a delay line [28]. A x20 water immersion lens (W Plan Apochromat DIC VIS-IR) was used for image acquisition. At least 3 wide-field images were acquired per nerve and per condition. For each image, the number of myelin ovoids and the percentage of degenerated fibers were counted using ZEN software (Carl Zeiss Microscopy, Rueil Malmaison, France).

### 2.11. Cell Culture and Transfection

Mouse Schwann cells MSC80 (ExPASy, CVCL_S187) were grown in DMEM (Dulbecco’s modified Eagle’s medium) (Gibco Life Technologies, Thermo Fischer Scientific, France) supplemented with 2 mM L-glutamine, 100 U/mL^−1^ penicillin/streptomycin and 5% (*v*/*v*) heat-inactivated fetal bovine serum (all supplements were from Invitrogen). Cells were maintained at 37 °C in an atmosphere of 5% CO_2_ and were passaged when they were 80–90% confluent, twice a week. Five 15 cm dishes of 70–80% confluent mouse Schwann cells were separately transfected with 30 μg of pLKO.1-puro vector containing five commercial VDAC1 shRNA (Sigma-Aldrich, Saint-Quentin-Fallavier Cedex, France, sh1#TRCN0000012388, sh2#TRCN0000012389, sh3#TRCN0000012390, sh4#TRCN0000012391 and sh5#TRCN0000012392) using 80 μL of Lipofectamine 2000 (Invitrogen) and 1,5 mL of Opti-Mem (Gibco Life Technologies, Thermo Fischer Scientific, France). After 7 h, the medium was changed to a fresh complete culture medium enriched with 2 mM of glutamine. Following 48 h after transfection, cells were treated with 0.5 μg/mL of Puromycin (Gibco Life Technologies, Thermo Fischer Scientific, France) for selection. Antibiotic was maintained in cell medium for one week, then cells were collected for Western blot.

### 2.12. Protein Extraction and Western Blotting

Transfected cells were washed in PBS, lysed in lysis buffer (10 mM Tris, pH 7.4, 150 mM NaCl, 1% Triton X-100, 0.1% SDS, 0.5% sodium-deoxycholate, 1 mM EDTA, 50 mM NaF, 1 mM NaVO4, protease inhibitor cocktail [Sigma-Aldrich]) for 15 min on ice, and centrifuged at 14,000 rpm at 4 °C to pellet cell debris. Sciatic nerves were dissected from eight-week-old mice with or without crush, washed in PBS and directly fixed with 4% of PFA for 10 min. After removal of the epineurium and perineurium, the nerves were homogenized by sonication in lysis buffer. Cellular debris was removed by centrifugation at 13,000 g for 5 min at 4 °C and protein was quantified by the bicinchoninic acid method using bovine serum albumin as a standard. Then, samples were denatured at 98 °C, loaded on 10% SDS-PAGE, and transferred on PVDF membranes for immunoblotting. Antibody against the phospho-S63-cJun (1/100, BD Biosciences, Ref. 558036, Le Pont de Claix Cedex, France) was from mouse and antibodies against VDAC (1/1000, Cell Signaling, Ref. 4866, Ozyme, France), phospho-Thr183/Tyr185-SAPK/JNK (1/1000, Cell Signaling, Ref. 9251, Ozyme, France), phospho-Thr202/Tyr204-p44/42 MAPK (ERK 1/2) (1/1000, Cell Signaling, Ref. 9101, Ozyme, France), phospho-Thr180/Tyr182-p38 (1/1000, Cell Signaling, Ref. 9211, Ozyme, France), Cleaved Caspase-3 (1/1000, Cell Signaling, Ref. 9661, Ozyme, France), total non-phosphorylated JNK (1/1000, Cell Signaling, Ref. 9252, Ozyme, France) were from rabbit. Antibody against phospho-S87-bcl2 was from goat (1/500, Santa Cruz Biotechnology, Ref. sc-16323, Dallas, TX, USA).

### 2.13. Validation of Fluorescent Probes and Anesthesia Control 

Mouse sciatic nerves expressing mito-GCaMP2 or mito-SypHer were isolated three weeks after AAV particles infection. Nerves were washed in PBS and incubated in Leibovitz’s L15 medium (Gibco Life Technologies, Thermo Fischer Scientific, Nimes, France) for 3 h at 37 °C in an atmosphere of 5% CO_2_. Sciatic nerves infected with mito-SypHer probe were treated separately with sodium azide 3 mM solution at pH 3.2 (Sigma-Aldrich, Ref. S2002, Saint-Quentin-Fallavier Cedex, France) or with ammonium chloride 30 mM solution at pH 8 (Sigma-Aldrich, Ref. A4514, Saint-Quentin-Fallavier Cedex, France) for 5 min. Sciatic nerves infected with mito-GCaMP2 were treated separately with calcium chelator EDTA 1 mM solution (Sigma-Aldrich, Ref. ED255, Saint-Quentin-Fallavier Cedex, France) or calcium chloride 100 µM solution (Sigma-Aldrich, Ref. S3014, Saint-Quentin-Fallavier Cedex, France) and saponine 20 µg/µL solution (Sigma-Aldrich, Ref. S4521, Saint-Quentin-Fallavier Cedex, France) for 5 min. Mito-SypHer and mito-GCaMP2 probe intensities were quantified at 985 nm using multiphoton microscope. 

### 2.14. Electron Microscopy and Morphometry

Mouse sciatic nerves were fixed for 20 min in situ with 4% PFA and 2.5% glutaraldehyde in 0.1 M phosphate buffer (pH 7.3), then removed and post-fixed overnight in the same buffer. After washing 30 min in 0.2 M PBS phosphate buffer, samples were incubated with 2% osmic acid in 0.1 M phosphate buffer for 90 min at RT. They were then washed in 0.2 M PBS phosphate buffer, dehydrated using ethanol gradient solutions and embedded in epoxy resin. To obtain electron microscopy images, ultrathin (70 nm) cross-sections were cut and stained with 1% uranylacetate solution and lead-citrate and analyzed using a HITACHI H7100 electron microscope. Semi-thin cross-sections (0.7 μm) were cut using a microtome (Leica, RM 2155, Wetzlar, Germany) with a diamond knife (Histo HI 4317, Diatome, Hatfield, UK). Sections were stained with blue from toluidine and imaged using an AxioScan slide scanner (Zeiss, Nantes, France). G-ratio was determined using the GRatioCalculator plugin of Image J. At least 200 fibers were analysed per animal.

### 2.15. Electrophysiology

Standard electroneurography was performed on mice anesthetized with 3% isoflurane. A pair of steel needle electrodes (AD Instruments, MLA1302, Oxford, UK) was placed subcutaneously on the back of the upper thigh of the animal at the sciatic nerve notch (proximal stimulation). A second pair of electrodes was placed on the leg along the tibial nerve above the ankle (distal stimulation). Supramaximal square wave pulses lasting 10 ms and 1 mA for mice were delivered using a PowerLab 26T (AD Instruments, Oxford, UK). Compound muscle action potential (CMAP) was recorded from the intrinsic foot muscles using steel electrodes. Both amplitudes and latencies of CMAP were determined. The distance between the two sites of stimulation was measured alongside the skin surface with fully extended legs, and nerve conduction velocities (NCVs) were calculated from sciatic nerve latency measurements.

### 2.16. Data and Statistical Analysis

Data are represented as mean ±SEM or ±SD according to the sample population size. The size of the animal groups was determined using BioStatTGV (https://biostatgv.sentiweb.fr, accessed on 30 June 2018). Statistical significances were determined using a two-tailed Student’s *t* test, one-way ANOVA followed by a Dunnett’s multiple comparison *post hoc* test and two-way ANOVA followed by a Sidak’s multiple comparison *post hoc* test. Significance was set at * *p* < 0.05, ** *p* < 0.01, or *** *p* < 0.001. ns indicates non-significant differences (*p* > 0.05). *n* indicates the number of independent experiments.

## 3. Results

### 3.1. Live Imaging of mSC Mitochondria and Cytoplasm and Fluorescent Probes Validation

The right sciatic nerve of two-month-old mice was injected with a solution of adenovirus or adeno-associated virus (AAV) expressing a fluorescent probe in order to transduce mSC in vivo [25,26,29]. Three weeks later, mice were anesthetized, their sciatic nerve exposed and placed under a multiphoton microscope allowing mSC time-lapse imaging in physiological conditions (Figure 1a) [26]. We used the fluorescent probe mito-Dsred2 [26] to measure mitochondrial size and movements, mito-Sypher [30] for mitochondrial matrix pH, mito-GCaMP2 [31] for mitochondrial matrix calcium content and cyto-GCaMP2 to assess of cytoplasmic calcium content. Mito-Dsred2 probe had previously been validated in vivo [26]. Mito-GCaMP2 and cyto-GCaMP2 were validated by incubating transduced nerves in successive baths of EDTA, a calcium chelator, and CaCl2, a calcium donor, in a mild saponin detergent (Figure 1b,c). Mito-Sypher was validated by incubating infected nerves in successive baths of sodium azide and ammonium chloride (Figure 1d), which are, respectively, the inhibitor and activator of the probe activity [32]. While the decrease of fluorescence with sodium azide was slightly beyond significance, we assumed that probes were functional in mSC in vivo and they did not induce detectable abnormalities or demyelination. 

### 3.2. Sciatic Nerve Injury Induces Mitochondrial Calcium Release in the Cytoplasm, Mitochondria Slowdown and Matrix Alkalization in mSC

Next, anesthetized animals were placed under the microscope and the probe fluorescence emitted by mSC in their sciatic nerves was recorded for around 20 min to measure baseline mitochondria velocity and fluorescence intensity. Then, nerves were injured through carefully crushing around 5 mm upstream of the imaging area in order to induce traumatic demyelination [3,27,33]. Changes occurring in probe-labelled mSC mitochondria downstream of the crush were recorded every 20 min over 5 h. Mito-Dsred2 probe reported a slow but significant decrease of the mitochondrial speed starting 120 min after the injury (Figure 2a). Looking at the mitochondrial length, we classified them as short (0.1 to 1 μm), medium (1 to 3 μm) and long (>3 μm)(Figure 2b) accordingly to their respective average speed in mSC [26]. We did not observe any significant change in the frequency of each size category (Figure 2b), suggesting that no fragmentation occurred. This was the opposite of what occurred after animal death as previously reported [26], showing that other mechanisms were involved. Mito-Sypher reported a significant and persistent increase in pH starting 120 min after the injury (Figure 2c), showing an alkalization of the mitochondrial matrix. 

Mito-GCaMP2 fluorescence showed a sharp decrease of mitochondrial calcium around one hour after the injury (Figure 3a). This loss of calcium from mitochondria timely correlated with a surge of calcium in the cytoplasm of mSC reported by cyto-GCaMP2 (Figure 3b). This clear time correlation indicated that mitochondrial calcium was released in the cytoplasm. This was just a pulsed release, as mitochondrial and cytosolic calcium levels quickly returned to their basal initial values. However, mitochondrial calcium significantly increased 2 h after the injury, indicating a late hypercalcemia of the mitochondrial matrix. Taken together, mitochondrial calcium release to the cytoplasm occurred before the increase in matrix pH and the slowdown of mitochondria movements (Appendix A). Matrix hypercalciema then occurred following this change in pH (Appendix A). 

### 3.3. VDAC1 and mPTP Are Responsible for Mitochondrial Calcium Release in the Cytoplasm

After triggering traumatic demyelination, mitochondrial calcium dynamics changed quickly regarding the overall process [3] both in mitochondria and in cytoplasm. As mitochondrial calcium release represents a signaling event in cells, we next investigated the molecular mechanisms that mediate this release. VDAC1 is a porin ion channel [34] highly expressed in the outer mitochondrial membrane of mSC [35] and regulating mitochondrial calcium release, cell signaling, apoptosis and dedifferentiation [36,37]. We characterized two small hairpin inhibitory RNAs (shRNA) selectively silencing mouse VDAC1 expression in Schwann cell line MSC80 (Figure 4a) and in mSC in vivo (Figure 4b). A control shRNA with no target in mammalian cells was also characterized. We transduced mSC in vivo with viral vectors expressing both the fluorescent probes and the shRNAs. We observed that both shRNAs targeting VDAC1 significantly lowered the mitochondrial calcium decrease and the concomitant cytoplasmic calcium increase following nerve injury, while the control shRNA had no effect (Figure 4c). This indicated that VDAC1 expression was required for the efflux of calcium from the matrix to the cytoplasm. In order to confirm this, we also tested TRO19622 (Olesoxime), a neuroprotective drug that has been shown to bind VDAC [38,39], as an inhibitor of calcium release through VDAC1. TRO19622 inhibited both mitochondrial calcium decrease and cytoplasmic calcium increase in mSC following injury (Figure 4d). Hexokinase is known as the main endogenous inhibitor of VDAC through a well-characterized direct interaction [40]. Indeed, the dissociation of hexokinase from VDAC through Methyl Jasmonate (MJ) binding to HexoKinase (HK) induces several mitochondrial and cellular modifications [41]. When nerves expressing mito-GCaMP2 were treated for two hours with MJ in vivo, mitochondrial calcium levels dropped significantly (Figure 4e) in mSC, indicating that dissociating HK from VDAC was sufficient to release calcium even in the absence of injury.

However, as VDAC is located in the outer mitochondrial membrane, the involvement of another channel crossing the inner mitochondrial membrane is required to allow the release of matrix calcium in the cytoplasm. VDAC participates in the formation of the mPTP, which crosses both the inner and outer mitochondrial membrane [42]. To check the involvement of mPTP in the release of calcium following nerve injury, we selectively blocked mPTP activity using cyclosporine A [43]. The concentration required to block mPTP was determined using a saturating concentration of auranofin, a drug that opens mPTP [44], combined with an increasing concentration of cyclosporine A (Appendix A). At a blocking concentration, cyclosporine A decreased the release of mitochondrial calcium following nerve injury (Figure 5), showing that mPTP is indeed involved in this process in addition to VDAC. 

### 3.4. The Release of Mitochondrial Calcium via VDAC1 Induces Schwann Cell Demyelination In Vivo via MAPK and CJUN ACTIVATION

We next investigated whether the release of mitochondrial calcium following injury could activate pathways known to be involved in the demyelination process: ERK1/2, p38, JNK [45,46,47] and cJun phosphorylation and cJun relocalization in the nucleus [48]. Phospho-ERK1/2, phospho-p38, phospho-JNK and phospho-cJun significantly increased 4 and 12 h after sciatic nerve injury and blocking VDAC through TRO19622 treatment strongly reduced the activation of these pathways (Figure 6a and Appendix A). Conversely, MJ treatment spontaneously activated these pathways in non-injured nerves (Figure 6a and Appendix A). Additionally, we observed a strong reduction of the nuclear localization of phospho-cJun when VDAC1 was silenced (Figure 6b) or blocked through TRO19622 (Figure 6c). The opening of VDAC through MJ treatment induced phospho-cJun nuclear enrichment in the absence of injury (Figure 6d). Taken together, this shows that VDAC opening induces the activation of the demyelination pathways in sciatic nerves and the enrichment of phospho-cJun in the nucleus of mSC.

To go further, we investigated demyelination using typical defects induced in the cellular morphology of mSC as described previously [49]. mSC expressing GFP have homogeneously large and long morphology in vivo, and, after demyelination, these cells acquire non-myelin forming cells features characterized by a heterogeneous thin and split morphology with fine extensions (Figure 6e). We used these striking morphological changes to count mSC and non-myelin-forming cells in injured and non-injured nerves expressing GFP and control or VDAC1 shRNAs. Nerve injury induced demyelination features in a large majority of cells expressing control shRNA, while significantly fewer cells showed a demyelination phenotype when they expressed VDAC1 shRNAs (Figure 6f), showing that VDAC1 silencing protected cells from demyelination. 

We also tested the pro-demyelinating effect of MJ and the anti-demyelinating effect of TRO19622 in vivo using coherent anti-stokes Raman scattering (CARS) imaging of myelin as read-out for demyelination. CARS is a non-linear microscopy that allows imaging lipids, hence myelin, in vivo or ex vivo without staining [27,50]. Two-month-old mice were treated with 3 mg/kg of TRO19622 or vehicle subcutaneously for five consecutive days. On the second day, these mice were anesthetized, and their right sciatic nerves were exposed and injured. Four days later, injured nerves were collected and freshly analyzed using CARS imaging of myelin. In injured nerves treated with vehicle, the myelin sheath was fragmented in ovoids (Figure 6g), which is a typical morphological feature of demyelination [3,4]. This fragmentation was quantified by counting the number of ovoids or by counting the percentage of demyelinating fibers (Figure 6h). When animals were treated with TRO19622 before and during injury, this fragmentation was absent or very preliminary (Figure 6g, quantified in 6 h), showing that inhibiting mitochondrial calcium-release prevented traumatic demyelination in vivo. In addition, when nerves were treated with MJ in absence of injury, a spontaneous demyelination occurred (Figure 6g,h), suggesting that VDAC opening was sufficient for the initiation of demyelination.

### 3.5. Basal Calcium Levels and Mitochondrial Calcium Release Kinetics Are Altered in db/db Diabetic Mice

Diabetes represents a significant cause of demyelinating peripheral neuropathy worldwide. However, molecular and cellular mechanisms underlying demyelination in this condition remain unclear. As diabetes alters metabolism and mitochondrial functions [22,51], we sought to investigate the demyelination process at the cellular and subcellular level in the *db/db* mouse line, a well-accepted mouse model of diabetes. This model displays peripheral nerve defects (thinner myelin, reduced nerve conduction velocity [NCV], motor impairment) and activation of demyelination signaling at early adulthood [52,53,54]. However, peripheral nerve demyelination is not significant before 1 to 1.5 years [52,53], suggesting that cellular dysfunctions may exist at an early age in mSC, but demyelination is slowly progressive. 

We first investigated the basal levels of mitochondrial and cytoplasmic calcium using the previously described expression of mito- or cyto-GCaMP2 probes in mSC in vivo. These cells had significantly less calcium in the mitochondria (Figure 7a) and more calcium in cytoplasm (Figure 7b) in homozygous *db/db* mice than in heterozygous control mice, suggesting that mitochondrial calcium homeostasis is altered in the mSC of diabetic mice. To investigate further, we crushed the nerves and followed mitochondrial and cytoplasmic calcium dynamics in real time in vivo. In order to compare changes between the two genotypes, data were normalized on basal values. While in heterozygous control mice, crush induced a similar mitochondrial calcium release in the cytoplasm as seen in wild-type, homozygous diabetic mice displayed a significantly sustained mitochondrial calcium release both seen with mito- and cyto-GCaMP2 probes (Figure 7c). This suggested that VDAC permeability to calcium is altered in the mSC of diabetic nerves. Taken together, we propose that VDAC1 is leaky in the mSC of diabetic nerves. This results in a higher concentration of calcium in the cytoplasm and in a cell prone to demyelination. 

### 3.6. Blocking Mitochondrial Calcium through VDAC Reduces Peripheral Nerve Defects and Peripheral Neuropathy in db/db Diabetic Mice

To check this hypothesis further, we treated *db/db* diabetic mice and controls with a daily dose of TRO19622 for 15 days or 30 days and investigated the molecular and cellular defects occurring in mSC and sciatic nerves of these mice. Firstly, we analyzed the nuclear distribution of phospho-cJun in mSC as described previously. More phospho-cJun was found in the nucleus of non-treated *db/db* mice versus heterozygous control and TRO19622 treatment significantly reduced this amount (Figure 7d). This indicated that mSC are indeed in a demyelinating status in *db/db* mice and TRO19622 prevented or reversed this status. 

To check the impact of diabetes and TRO19622 treatment on the myelin sheath, we analyzed the G-ratio (axon diameter/fiber diameter) using semi-thin electron microscopy sections. As reported previously [52,53,54], diabetes significantly reduced myelin thickness in *db/db* mice (Figure 7e), which translated as a higher G-ratio (Figure 7f). The treatment of *db/db* mice with TRO19622 for one month decreased the G-ratio toward control heterozygous mice (Figure 7f). 

We also investigated the effect of diabetes and TRO19622 treatment on *db/db* mouse NCV, which is related to the amount of myelin covering axons. As previously described [52,53,54], NCV was significantly reduced in *db/db* mice vs. control heterozygous mice before treatment (Figure 7g left panel), indicating axon potential propagation was delayed by defective myelin. TRO19622 treatment increased NCV in *db/db* mice to control heterozygous treated mice values (Figure 7g right panel). This was consistent with the positive effect of the drug on the myelin sheath thickness. 

Finally, we investigated the mouse motor behavior using Rotarod and grip tests. Animals were treated daily with the drug or vehicle for 4 weeks starting at the age of 1 month. These animals were tested 1 week and 4 weeks after treatment started. We found that muscular strength was affected in diabetic mice and a 4-week treatment with TRO19622 reversed this defect (Figure 7h). Diabetic mice also performed more poorly on the rotarod than control heterozygous mice, and treatment with TRO19622 significantly improved their performances vs. non-treated *db/db* mice after a 4-week treatment (Figure 7i). We then stopped the treatment and tested animals again 8 weeks later. In these conditions, the observed benefit of TRO19622 treatment was lost (Figure 7i left panel).

Taken together, these data indicated that blocking mitochondrial calcium leak in *db/db* mSC through TRO19622 treatment allowed for preventing or reversing the molecular, cellular, physiological and behavioral defects occurring in these mice. Treatment had to be sustained in order to maintain the benefit.

## 4. Discussion

While demyelinating peripheral nerve diseases include a large spectrum of acquired and inherited disabling diseases, the mechanisms of SC demyelination remain elusive. Peripheral nerve demyelination does not result from cell death but from myelinating SC dedifferentiation [55]. So, demyelination is a cellular program in which the myelinating SC enters upon specific signals, such as axonal injury in a trauma. These triggering signals are transduced in the cell and drive the activation of MAPK demyelination pathways (phospho-ERK1/2, phospho-P38 and phospho-JNK activation) followed by the recruitment of phosphorylated cJun in the nucleus. Here we describe an earlier step, the release of mitochondrial calcium and changes in mitochondria physiology.

Using a viral approach to express fluorescent probes in the mitochondria of myelinating SC in the sciatic nerve of living mice and a multiphoton microscope for time-lapse live imaging, we show that the release of mitochondrial calcium occurs as soon as one hour after inducing demyelination by nerve injury. This release is followed, in this specific order, by a burst of calcium in the cytoplasm, the slowing of mitochondrial movements, the increase of mitochondrial pH and mitochondrial hypercalcemia (Appendix A). While the release of calcium in the cytoplasm is a constant feature following the triggering of demyelination in mSC, the hypercalcemia is not always occurring as illustrated in Figure 7c. We believe this may be due to the physiological state of the cells in some animals, but we have no idea of the cause.

Using silencing and activation/inhibition with drugs, we show that VDAC1 is responsible for the release of mitochondrial calcium in the cytoplasm. This pulse of mitochondrial calcium through VDAC1 activates the known demyelination pathways ERK1/2, p38 and JNK, leading to cJun phosphorylation in the nucleus, which characterizes the demyelination program in myelinating SC [48]. However, several other unclear cellular processes are also engaged, and this will result in the collapse of the cell structure, the breakdown of the myelin [3,4] and the recruitment of macrophages to help to clear myelin debris [4]. Once demyelination is completed, around 5 days after the crushing of the sciatic nerve, then dedifferentiated SC will be able to remyelinate axons. 

The release of calcium by the mitochondria is one of the earliest steps recorded after nerve injury, and our data indicate that this step is necessary and sufficient for triggering the demyelination program. How this burst of mitochondrial calcium through VDAC1 channels activates ERK1/2, p38 and JNK pathways is not clear. However, mitochondrial calcium release is an essential cell signal triggering death and differentiation [56]. Moreover, cytoplasmic calcium stimulates ERK1/2 [57] and JNK [58] activity. Several studies also reported an association of ERK1/2, p38 and JNK and their direct activators such as Raf, Sab, and MKK4 with mitochondria [59]. So, it is likely that mitochondrial calcium release in the cytoplasm directly activates MAPK demyelination pathways, and these cascades pathways propagate the demyelination signal all over the cell. Nevertheless, other mechanisms cannot be excluded, such as the production of reactive oxygen species (ROS) in calcium-activated mitochondria to stimulate ERK1/2, p38, and JNK pathways [59,60]. Whatever the precise mechanism, this mitochondrial signaling after nerve injury never reached the cell death level as cell-death-related pathways such as caspase 3 and Bcl2 were not activated. Mitochondrial signaling is therefore able to generate a differentiation process instead of cell death, as previously reported in myoblasts [61]. 

VDAC has numerous binding partners that control its permeability, in particular HK. HK binding to VDAC reduces the permeability of the pore notably to calcium [36]. In SC methyl jasmonate, a compound that uncouples HK from VDAC [41], induced mitochondrial calcium release and demyelination showing that HK is essential to control demyelination. Indeed, mutation in the HK gene is responsible for demyelinating Charcot–Marie–Tooth (CMT) 4G disease [62]. In contrast, we show that TRO19622, a compound that binds to VDAC [39], blocks mitochondrial calcium release through VDAC, therefore preventing demyelination. 

Our live-imaging data suggested that mitochondrial and cytoplasmic calcium homeostasis is altered in the mSC of diabetic mice. We linked it to a change in kinetics of mitochondrial calcium release following nerve injury, suggesting VDAC permeability to calcium is increased in mSC of diabetic mice. This could result in a sustained leak of mitochondrial calcium in the cytoplasm and a higher probability to engage the demyelination program is diabetic mice. This was confirmed using TRO19622 to inhibit calcium release through VDAC in diabetic mice: treated mice showed a reduction of the demyelination marker in nerves, an improved myelin thickness, an improved nerve conduction velocity and higher performances on motor behavioral tests. The higher permeability of VDAC to calcium and the highest inclination of mSC to demyelinate may explain the chronic demyelination that occurs in diabetic patients’ nerves. Therefore, these results suggest that TRO19622, but also other inhibitors of VDAC permeability to calcium, may constitute an opportunity to treat diabetic peripheral neuropathy. Nevertheless, the benefit of blocking calcium release through VDAC remains to be confirmed in another animal model, as diabetic mice only show a subtle alteration of peripheral nerve functions. 

Mitochondrial calcium release through VDAC has other consequences that may also contribute to the demyelination program. First, the slowdown of mitochondrial motility we observed after nerve crush is likely to be due to the increase of cytoplasmic calcium around the mitochondria. Indeed, calcium alters the activity of molecular motors that mediate mitochondrial movements in the cell [63]. Second, mitochondrial pH is linked to cytoplasmic calcium concentration [64]. So, when mitochondria release calcium, their pH changes. However, while hypocalcemia is transitory, the pH changes in the long term (at least for 5 h), suggesting that mitochondrial respiratory activity is increased [65]. As ATP controls the uptake of calcium from the endoplasmic reticulum [66], the mitochondrial hypercalcemia occurring 4 h after nerve crush may also reflect an increase in respiratory activity. Taken together, this suggests mitochondrial calcium release after nerve crush profoundly changes mitochondria physiology in mSC, resulting in increased mitochondrial activity. How mitochondrial activity and metabolic changes participate in the demyelination program is an interesting open question.

## 5. Patents

The work reported in this manuscript resulted in a patent titled “Methods and pharmaceutical compositions for treating peripheral demyelinating diseases US10758548B2”.

## Figures and Tables

**Figure 1 biomedicines-10-01447-f001:**
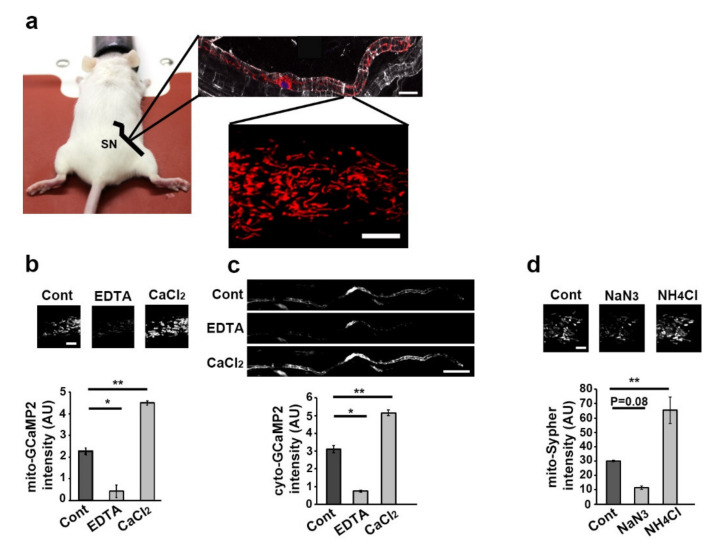
Live imaging technique and fluorescent probes validation (**a**)—Schematic representation of the imaging technique used in this work. The sciatic nerve (SN) of anesthetized mice illustrated by the black line was exposed under the lens of a multiphoton microscope. Myelinating SC, stained for E-cadherin (white), contains mitochondria labelled through adenovirus-delivered mito-Dsred2 expression (red) in particular around the cell nucleus labelled with TOPRO3 (blue) (scale bar= 10 µm). This technique allows the selective in vivo multiphoton imaging of mitochondria in some parts of living SC (scale bar = 5 µm. Same magnification for all pictures.). All mice were 8 to 11 weeks old. (**b**) Mouse sciatic nerves expressing mito-GCaMP2 were incubated in a bath containing Leibovitz’s L15 medium without (Cont) or with EDTA (1 mM) or calcium chloride (CaCl_2_, 100 µM). Probe fluorescence was imaged (upper panels pictures, scale bar = 5 µm. Same magnification for all pictures.) and measured (lower panel graph). (**c**) Mouse sciatic nerves expressing cyto-GCaMP2 were incubated in a bath containing Leibovitz’s L15 medium without (Cont) or with EDTA (1 mM) or calcium chloride (CaCl_2_, 100 µM). Probe fluorescence was imaged (upper panels pictures, scale bar = 50 µm. Same magnification for all pictures.) and measured (lower panel graph). (**d**) Mouse sciatic nerves expressing mito-Sypher were incubated in a bath containing Leibovitz’s L15 medium without (Cont) or with sodium azide (Na3, 3 mM pH 3.2) or ammonium chloride (NH4Cl, 30 mM pH 8). Probe fluorescence was imaged (upper panels pictures, scale bar = 5 µm. Same magnification for all pictures.) and measured (lower panel graph). Error bars show SEM. Statistical tests are one-way ANOVA comparing the control with each other condition. * *p* < 0.05, ** *p* < 0.01.

**Figure 2 biomedicines-10-01447-f002:**
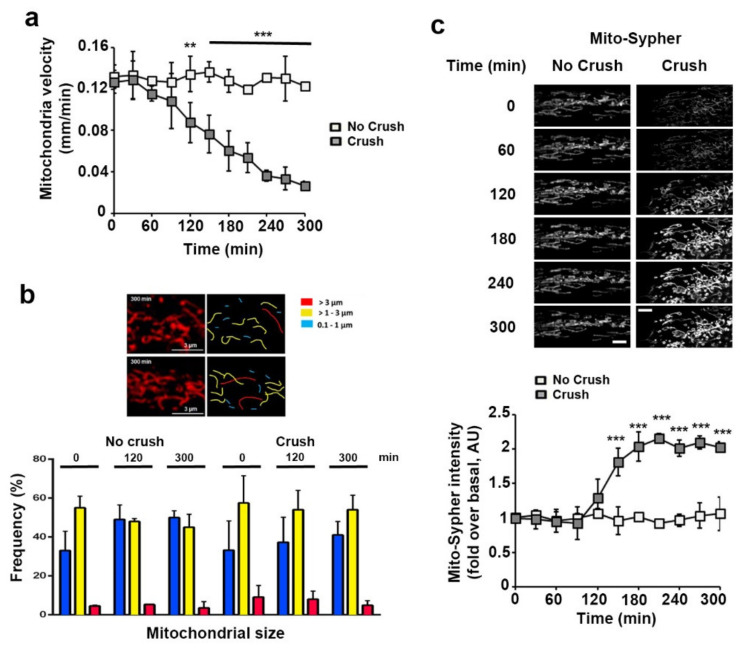
Mitochondrial mobility and pH change during Schwann cell demyelination. (**a**) Mitochondrial velocity in myelinating SC of control (No crush, *n* = 2 mice) and crushed (Crush, *n* = 5 mice) nerves was measured using adenovirus-delivered mito-DsRed2. Velocity is shown in mm traveled in one minute. Nerve injury occurs at t = 0. (**b**) **Upper panels**: Representative in vivo images of SC mitochondria (left panels) showing how mitochondrial length is characterized and quantified (right panels) in non-crushed and crushed nerves. **Lower panel:** Frequency histogram of mitochondrial size in control (No crush, *n* = 2 mice) and crushed (Crush, *n* = 5 mice) conditions at three successive time points. Scale bar = 3µm. Same magnification for all pictures. (**c**) **Upper panels:** Representative in vivo images of SC mitochondria labeled with adenovirus-delivered mito-Sypher probe in non-crushed (No Crush) and crushed nerves (Crush) at successive time points. Scale bar= 5 μm. Same magnification for all pictures. **Lower panel:** the average fluorescence intensity of the probe was measured for more than 100 mitochondria at the successive indicated time points in non-crushed (No Crush, *n* = 3 mice) and crushed nerves (Crush, *n* = 4 mice). The probe fluorescence intensity is normalized over the basal condition before crush. Error bars indicate SEM. Statistical tests are repeated measures two-way ANOVA Sidak *post hoc* test comparing non-crushed to crushed nerves values. All mice were 8 to 11 weeks old. ** *p* < 0.01, *** *p* < 0.001.

**Figure 3 biomedicines-10-01447-f003:**
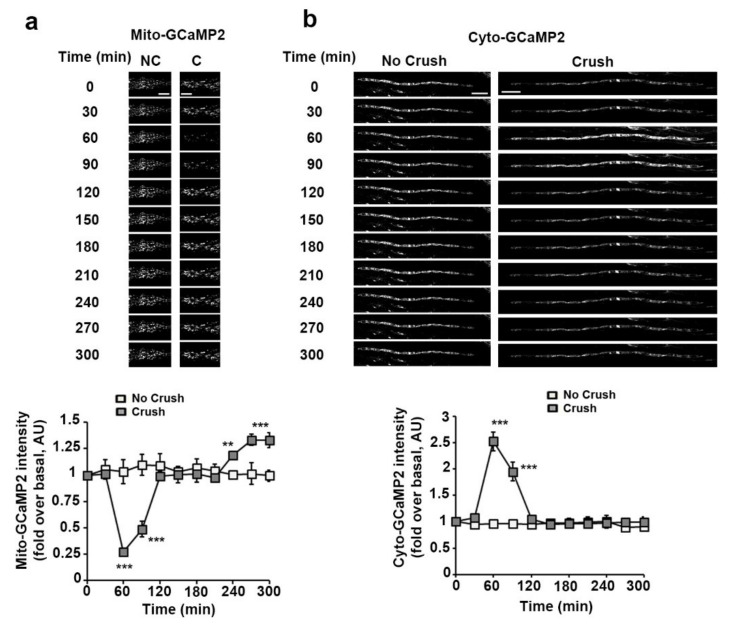
Mitochondrial and cytoplasmic calcium dynamics in SC following nerve injury. (**a**) Upper panels: Representative in vivo images of SC mitochondria labeled with adenovirus-delivered mito-GCaMP2 probe in non-crushed (No Crush) and crushed nerves (Crush) at successive time points. Scale bar= 5 μm. Same magnification for all pictures. **Lower panel:** the average fluorescence intensity of the probe was measured for more than 100 mitochondria at the successive indicated time points in non-crushed (No Crush, *n* = 3 mice) and crushed nerves (Crush, *n* = 5 mice). The probe fluorescence intensity is normalized over the basal condition before crush. (**b**) **Upper panels:** Representative in vivo images of SC mitochondria labeled with adenovirus-delivered cyto-GCaMP2 probe in non-crushed (No Crush) and crushed nerves (Crush) at successive time points. Scale bar= 50 μm. Same magnification for all pictures. **Lower panel:** the average fluorescence intensity of the probe was measured for more than 100 mitochondria at the successive indicated time points in non-crushed (No Crush, *n* = 3 mice) and crushed nerves (Crush, *n* = 4 mice). The probe fluorescence intensity is normalized over the basal condition before crush. Error bars indicate SEM. Statistical tests are two-way ANOVA Sidak *post hoc* test comparing non-crushed to crushed nerves values. All mice were 8 to 11 weeks old. ** *p* < 0.01, *** *p* < 0.001.

**Figure 4 biomedicines-10-01447-f004:**
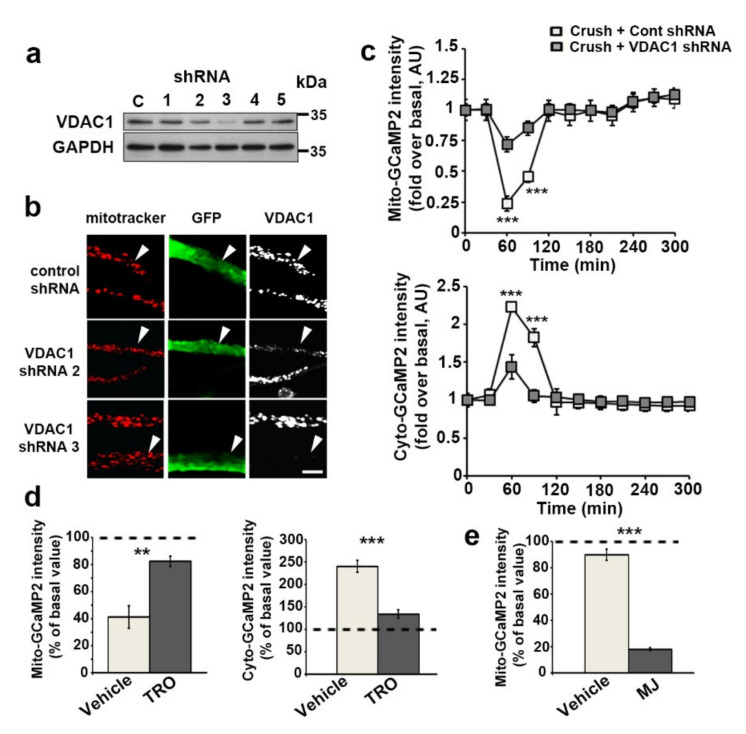
VDAC1 mediates mitochondrial calcium release during SC demyelination. (**a**) MSC80 SC were transfected with plasmids expressing puromycin and a control shRNA with no mammalian target (C) or 5 shRNAs (1–5) specifically targeting mouse VDAC1 under a U6 promoter. After a selection with puromycin, cells were lysed and their VDAC protein expressions were analyzed by WB. GAPDH is used as loading control. (**b**) AAV virus expressing VDAC1 shRNAs 2 or 3 in addition to GFP were injected in the sciatic nerve of mice. Three weeks later animals were sacrificed and the expression of VDAC in infected SC was analyzed by immunostaining on teased fibers. Infected cells expressing control shRNA and GFP (green, arrowheads) show mitochondria (mitotracker, red) harboring VDAC (white) while cells expressing VDAC1 shRNAs show reduced expression of VDAC. Scale bar = 5 μm. Same magnification for all pictures. (**c**) AAV virus expressing VDAC1 shRNAs 2 or 3 in addition to mito- or cyto-GCaMP2o was injected in the sciatic nerve of mice. The average fluorescence intensity of the mito-GCaMP2 (upper panel) and cyto-GCaMP2 (lower panel) probes was measured for more than 100 mitochondria at the successive indicated time points in non-crushed (No Crush, *n* = 3 mice) and crushed nerves (Crush, *n* = 4–5 mice). The probe fluorescence intensity is normalized over the basal condition before crush. Error bars indicate SEM. Statistical tests are two-way ANOVA Sidak post hoc test comparing crushed with control shRNA to crushed with VDAC1 shRNAs conditions. All mice were 8 to 11 weeks old. (**d**) Sciatic nerves transduced with viruses expressing mito-GCaMP2 (left panel, *n* = 3 mice and >100 mitochondria for both TRO and vehicle) or cyto-GCaMP2 (right panel, *n* = 3 and > 100 mitochondria mice for both TRO and vehicle) were injected with 2 μL of 20 μM TRO19622 (TRO) solution or vehicle 30 min before nerve injury and then live imaging. Probes’ fluorescence intensities were measured before injury and then 60 min after injury at the peak of mitochondrial calcium release. Values at t = 60 min are presented in percentage of basal values before crush. The dotted lines indicate 100% of the basal values. Error bars indicate SEM. Statistical tests are two-tailed Student’s *t*-test. All mice were 8 to 11 weeks old. (**e**) Sciatic nerves transduced with viruses expressing mito-GCaMP2 (left panel, *n* = 89 and 109 mitochondria for vehicle and MJ respectively, in four mice each) were injected with 1 μL of 57 μM methyl jasmonate (MJ) solution or vehicle 10 min before live imaging. No crush was performed and fluorescent mitochondria were imaged again 2 h later. Values at t = 2 h are presented in percentage values at t = 10 min. The dotted lines indicate 100% of the basal values. Error bars indicate SEM. Statistical tests are two-tailed Student’s *t*-test. All mice were 8 to 11 weeks old. ** *p* < 0.01, *** *p* < 0.001.

**Figure 5 biomedicines-10-01447-f005:**
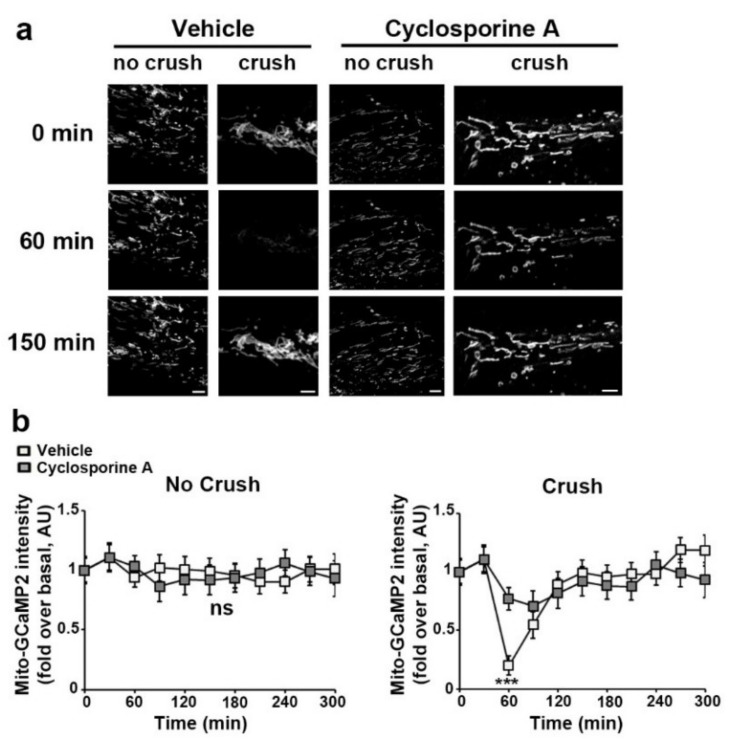
Blocking mPTP with cyclosporine A prevents mitochondrial calcium release in mSC following nerve injury. (**a**) Representative images of mSC mitochondria labeled with adenovirus-delivered mito-GCaMP2 probe in non-crushed (No Crush) and crushed (Crush) nerves at successive time points following treatment with 500 µm cyclosporine A (right panels) or vehicle (left panels). Scale bar= 3 μm. Same magnification for all pictures. (**b**) **Left panel**: Average fluorescence intensity of mito-GCaMP2 was measured for more than 100 mitochondria at successive time points in non-crushed nerves treated with cyclosporine A or vehicle. **Right panel**: Average fluorescence intensity of mito-GCaMP2 was measured for more than 100 mitochondria at the successive time points in crushed nerves treated with cyclosporine A or vehicle. The probe fluorescence intensity is normalized over the basal condition before crush or at the first imaging time point for non-crushed nerves. Error bars indicate SEM. Statistical tests are two-way ANOVA Sidak’s *post hoc* test comparing cyclosporine A and vehicle treated conditions. *n* = 3 animals for each condition. All mice were 8 to 11 weeks old. *** *p* < 0.001.

**Figure 6 biomedicines-10-01447-f006:**
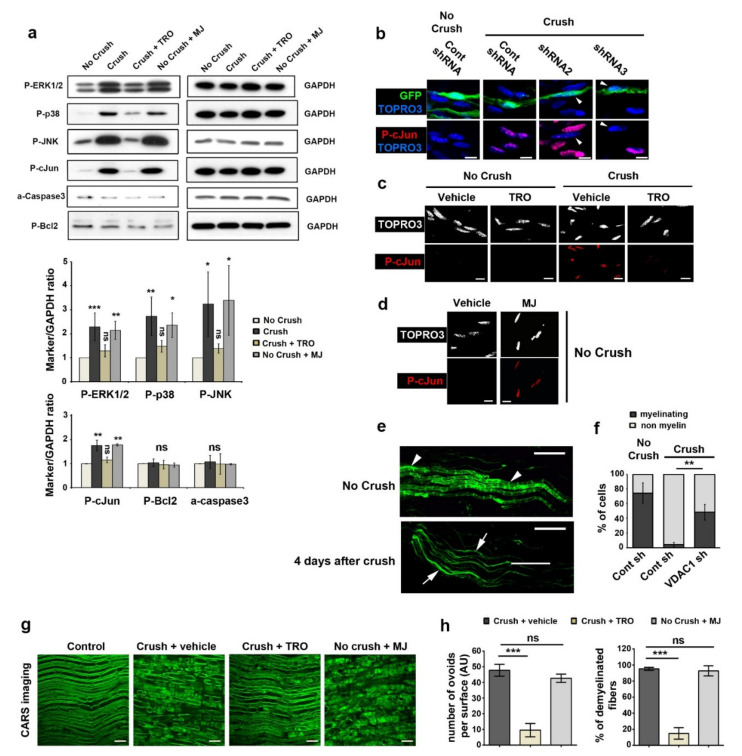
Demyelination is decreased after treatment with TRO19622 and spontaneously induced after treatment with MJ. (**a**) **Upper panels:** Western blot analysis for phospho-ERK1/2 (P-ERK1/2), phospho-p38 (P-p38), phospho-JNK (P-JNK), phospho-cJun (P-cJun), activated cleaved-caspase3 (a-caspase3) and phospho-Bcl2 (P-Bcl2) in the sciatic nerve of mice without injury (No crush), 4 h after injury (Crush), 4 h after injury with TRO19622 treatment 30 min before injury (Crush + TRO), or without injury but 4 h after methyl jasmonate treatment (No crush + MJ). GAPDH was used as loading control. Blots were stripped and re-hybridized with different antibodies against proteins of interest, so GAPDH is a loading control for different immunoblots. *n* = 3 independent experiments. **Lower panels:** WB results analyzed by densitometry and normalized on GAPDH respective values on the same blot. As similar changes were observed at 4 and 12 h after injury (see Appendix A for 12 h representative images), data from both experiments were pooled except for P-cJun, for which changes were consistent but much higher at 12 h. So, only 4 h data are presented. All data are presented as fold over No crush condition. Error bars show SD. Statistical tests are one-way ANOVA followed by a Dunnett’s multiple comparison *post hoc* tests for each marker. *n* = 2 (P-cJun) to 4 mice. ns= non-significant. (**b**) Immunohistochemistry for phospho-cJun (P-cJun, red) and nuclear TOPRO3 (blue) on teased fibers of mouse nerve transduced with virus expressing control shRNA or shRNA 2 or 3 targeting VDAC1 in addition to GFP (green). Without injury (No crush) non infected cells or cells expressing control shRNA and GFP show no P-cJun in their nucleus. Four days after injury (Crush), non-infected cells and cells expressing control shRNA and GFP express P-cJun in their nucleus, while cells expressing shRNA2 or 3 and GFP show low amount of P-cJun in their nucleus (arrowheads). Scale bars = 10 μm. Same magnification for all pictures. (**c**)—Immunohistochemistry for phospho-cJun (P-cJun, red) and nuclear TOPRO3 (white) on teased fibers of mouse nerve without injury (No crush) or 4 days after injury (Crush). In both cases, animals were treated with TRO19622 (TRO) or vehicle, once intraperitoneally (3 mg/kg) 10 h before injury, once intrasciatically (2 µL 20 µM) 30 min before injury, and then intraperitoneally for 4 consecutive days. Scale bars = 10 μm. Same magnification for all pictures. (**d**) Immunohistochemistry for phospho-cJun (P-cJun, red) and nuclear TOPRO3 (white) on teased fibers of mouse nerve without injury (No crush) 4 days after treatment with methyl jasmonate (MJ) or vehicle. Scale bars = 10 μm. Same magnification for all pictures. (**e**) Representative images of the morphological features occurring in myelinating SC during demyelination 4 days after nerve injury. These cells express virally-delivered GFP (Green). Arrowheads show myelinating SC and arrows non-myelin-forming SC. Scale bars = 100 μm. (**f**) Quantification of myelinating and non-myelinating SC frequency in nerves transduced with AAV- expressing Control shRNA (Cont sh) or shRNAs 2 or 3 targeting VDAC1 (VDAC1 sh) in addition to GFP with (Crush) or without nerve injury (No crush). Nerves were collected 4 days after the nerve injury and GFP-positive cells were counted as myelinating or non-myelin forming as shown in Figure 6e. Data are represented as mean ±SD. Statistical significances were determined using a two-tailed Student’s *t*-test. *n*= 5 (No Crush, Cont sh), 4 (Crush, Cont sh) and 4 (Crush, VDAC1 sh) mice. (**g**) Representative CARS images of sciatic nerves freshly collected in non-injured control animals (Control), in animals 4 days after crush nerve injury and after vehicle (Crush + vehicle) or TRO19622 (Crush + TRO, 3 mg/kg) treatment or in animals 4 days after injection of MJ in the sciatic nerve (No Crush + MJ). Scale bars= 20 μm. (**h**) The number of ovoid per surface unit (left) and the percentage of demyelinated fibers (right) were measured on the CARS images. Data are presented as mean ± SEM. *n* = 6 (Crush + vehicle), 6 (Crush + TRO) and 4 (No Crush + MJ) animals. Statistical analysis shows one-way ANOVA Sidak’s *post hoc* test. * *p* < 0.05, ** *p* < 0.01, or *** *p* < 0.001.

**Figure 7 biomedicines-10-01447-f007:**
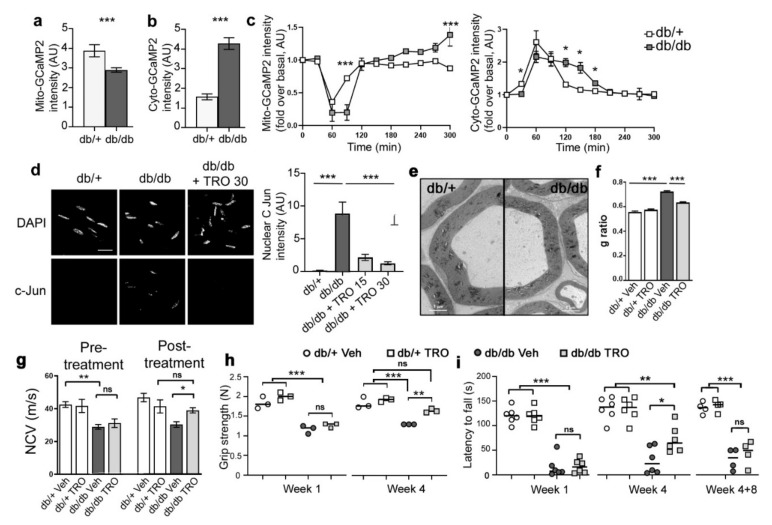
SCs of *db/db* diabetic mice show an altered mitochondrial calcium homeostasis and TRO19622 prevents motor, electrophysiological and cellular defects in these mice. Mitochondrial calcium (Mito-GCaMP2) (**a**), cytoplasmic calcium (Cyto-GCaMP2) (**b**) probe intensity in mSCs of control (*db/+*) and diabetic (*db/db*) mice in basal conditions. Mitochondrial calcium is decreased and cytoplasmic calcium increased in the mSCs of diabetic mice. Error bars indicate SEM. Statistical tests are unpaired Student’s *t*-test. (**a**) *n*= 77 (*db/+*), 168 (*db/db*) mitochondria (6 animals each genotype). (**b**) *n* = 11 (*db/+*), 9 (*db/db*) mSCs (3 mice each genotype). All mice were 8 to 11 weeks old. (**c**) Mitochondrial calcium (left) and cytoplasmic calcium (right) probe intensity in mSCs of control (*db/+*) and diabetic (*db/db*) mice following nerve crush (t = 0). Mitochondrial calcium release is significantly sustained at t = 90 min. The probe fluorescence intensity is normalized over the basal condition before crush. Error bars indicate SEM. Statistical tests are two-way ANOVA Sidak *post hoc* test. Left: *n* = 40 (*db/+*), 88 (*db/db*) mitochondria (3 mice for each genotype). Right: *n* = 7 (*db/+*), 5 (*db/db*) mSCs (3 mice for each genotype). All mice were 8 to 11 weeks old. (**d**) **Left panel:** Immunohistochemistry for phospho–c-JUN and DAPI nuclei staining on mouse sciatic nerve cryosections. Mice were treated with vehicle (db/+, *db/db*) or with TRO19622 (*db/db* + TRO 30, 3 mg/kg) for 30 days. Arrows indicate infected mSC nuclei. Scale bar: 50 μm. Same magnification for all pictures. **Right panel:** Quantification of nuclear phospho–c-JUN represented as fold over *db/+* mice. Noninfected neighbour cells were used as internal controls. Mice were treated with vehicle (db/+, *db/db*, 30 days) or with TRO19622 for 15 days (*db/db* + TRO 15) or 30 days (*db/db* + TRO 30). Error bars indicate SEM. Statistical tests are one-way ANOVA test. *n* = 28 (db/+), 17 (*db/db*), 17 (*db/db* + TRO 15), 19 (*db/db* + TRO 30) cells (3 mice of each condition). All mice were 8 to 11 weeks old. (**e**) Representative transmission electron micrograph images of sciatic nerve cross sections of *db/+* and *db/db* mice. Scale bar: 5 μm. (**f**) g ratio (axon diameter/fiber diameter) was measured on electron micrograph from *db/+* or *db/db* mice treated with vehicle (db/+ Veh, *db/db* Veh) or TRO16922 for 30 days (db/+ TRO, *db/db* TRO) (3 mg/kg). A minimum of 189 fibers was measured per animal. Statistical test is one-way ANOVA followed by a Dunnett’s multiple comparison post-hoc test. *n*= 3 mice for each condition. (**g**) Measure of sciatic nerve NCV of *db/+* and *db/db* mice before and after treatment with vehicle (db/+ Veh, *db/db* Veh) or TRO 19622 for 30 days (db/+ TRO, *db/db* TRO) (3 mg/kg). Measure of grip strength (**h**) and rotarod latency (**i**) of *db/+* and *db/db* mice at one week (Week 1) and four weeks (Week 4) of treatment with vehicle (db/+ Veh, *db/db* Veh) or TRO 19622 (db/+ TRO, *db/db* TRO) (3 mg/kg) and 8 weeks after stopping the 4-weeks treatment (Weeks 4+8). Data are expressed as the mean ± SEM. *n* = 12 mice for each group. Statistical test is one-way ANOVA followed by a Dunnett’s multiple comparison post-hoc test. * *p* < 0.05, ** *p* < 0.01, or *** *p* < 0.001.

## Data Availability

Not applicable.

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
