# Peer review of "Traumatic and Diabetic Schwann Cell Demyelination Is Triggered by a Transient Mitochondrial Calcium Release through Voltage Dependent Anion Channel 1"

_biomedicines, 2022, doi:10.3390/biomedicines10061447_

Round 1
Reviewer 1 Report
The article is devoted to the study of a transient mitochondrial calcium release in traumatic and diabetic Schwann cell demyelination.
Overall the paper is well-written, organised, the conclusions are logical and the finding of the authors that mitochondrial calcium release through Voltage Dependent Anion Channel could contribute to pathogenesis of demyelination seems to be important in the context of the search for new treatments for this disease.
However, there are some minor questions which must be addressed.
List of abbreviations used would benefit the paper.
Introductory section is well-written and contains enough information to understand the background and the goal of the research done.
References number 4 and 41 should be formatted according to MDPI requirements.
Materials and Methods section should include the information about db/db diabetic mice strain used in the research.
What are numbers1-5 in Figure 4a? It should be mentioned in figure legend.
What do dashed lines represent in Figure S2?
Figure 5 legend: “Blocking mPTP with cyclosporine A prevent mitochondrial …” The authors should check English language in this sentence as well as in some other parts of the manuscript. The concentration of cyclosporine A should be checked at page 12, line 27 (legend of Figure 5).
It would be nice to see the data (or additional information) on the specificity of antibodies used to determine phosphorylated versions of proteins (Phospho- 10 ERK1/2, phospho-p38, phospho-JNK and phospho-cJun). Do these antibodies interact with non-phosphorylated protein at certain background level?
How many times western-blots with samples shown on Figures 6a and Figure S3 were repeated?
How the authors would explain differences in kinetics at long times (longer than 200 minutes) seen in Figure 7 C (mitoGCAMP2) for db/+ strain with analogous experiment done in Figure 3a (where there is a rise of calcium after 200 min after crush)?
Page 17, lines 17, 21 - Figures S4 and S5 are absent in supplementary information.
Page 17, line 38 “Figure 7i left panel ” This statement should be clarified.
The authors should provide a justification that a required amount of animals was used for the experiments from the point of view of statistical analysis applied.
The paper can be accepted after minor revision.
Author Response
List of abbreviations used would benefit the paper.
We reviewed the guidelines of Biomedicines and we could not find any information on a list of abbreviations. Instead we found the recommendation to define abbreviations the first time they appear in the manuscript. So, we did this. Changes made in the manuscript are underlined in yellow.
References number 4 and 41 should be formatted according to MDPI requirements. OK Done
Materials and Methods section should include the information about db/db diabetic mice strain used in the research.
These precisions were added in the material and methods section: “Immunodeficient strain CB17/SCID mice (Janvier Labs, France) and db/db diabetic mouse strain (BKS.Cg-Dock7m+/+LeprdbJ Genetically Engineered Inbred, The Jackson Laboratory) were kept in the animal house facility of the Institute for Neurosciences of Montpellier in ventilated and clear plastic boxes, subjected to standard light cycles (12 h to 90 lux light, 12 h dark). db/db mice are used to model phase 1 to 3 of diabetes type II and obesity. Mice homozygous for the diabetes spontaneous mutation (Leprdb) demonstrate morbid obesity, chronic hyperglycemia, pancreatic beta cell atrophy and hypoinsulinemic.”
What are numbers1-5 in Figure 4a? It should be mentioned in figure legend.
We added the following to the figure legend: “MSC80 SC were transfected with plasmids expressing puromycin and a control shRNA with no mammalian target (C) or 5 shRNAs (1-5) specifically targeting mouse VDAC1 under a U6 promoter.”
What do dashed lines represent in Figure S2?
The Figure S2 title and legend was modified as following :
“Figure S2. 500 µM cyclosporine A blocks mPTP opening induced by 2µM auranofin in mSC in vivo
Mice expressing mito-GCaMP2 in mSC were anesthetized and mitochondrial calcium dynamic was analyzed through live two-photon imaging. Cyclosporine A (CsA) was injected in the nerve 30 minutes before imaging. Then, 2 µm auranofin was also injected in order to open mPTP and to release mitochondrial calcium. When concentration is too low (50, 100 and 200µM) CsA does not prevent mPTP opening with auranofin and mitoCGaMP2 probe fluorescence remains unchanged. Above 500µM CsA blocks mPTP opening despite auranofin inhibition.
Upper panel. Representative images of the mito-GCaMP2 probe fluorescence in the different conditions.
Lower panel. The mean mito-GCaMP2 fluorescence intensity was measured for more than 100 mitochondria at different time points for each conditions. n=2 mice. Dashed lines show the SD error bars for each condition and each time point.”
Figure 5 legend: “Blocking mPTP with cyclosporine A prevent mitochondrial …” The authors should check English language in this sentence as well as in some other parts of the manuscript. The concentration of cyclosporine A should be checked at page 12, line 27 (legend of Figure 5).
The Figure 5 title and legend was modified as following :
“Figure 5. Blocking mPTP with cyclosporine A prevents mitochondrial calcium release in mSC following nerve injury. (a)- Representative images of mSC mitochondria labeled with adenovirus-delivered mito-GCaMP2 probe in non-crushed (No Crush) and crushed (Crush) nerves at successive time points following treatment with 500 µm cyclosporine A (right panels) or vehicle (left panels). Scale bar= 3m. (b) Left panel: Average fluorescence intensity of mito-GCaMP2 was measured for more than 100 mitochondria at the successive time points in non-crushed nerves treated with cyclosporine A or vehicle. Right panel: Average fluorescence intensity of mito-GCaMP2 was measured for more than 100 mitochondria at the successive time points in crushed nerves treated with cyclosporine A or vehicle. The probe fluorescence intensity is normalized over the basal condition before crush or at the first imaging time point for non-crushed nerves. Error bars indicate SEM. Statistical tests are two-way ANOVA Sidak’s post hoc test comparing cyclosporine A and vehicle treated conditions. n=3 animals for each condition. All mice were 8 to 11 weeks old.”
It would be nice to see the data (or additional information) on the specificity of antibodies used to determine phosphorylated versions of proteins (Phospho- 10 ERK1/2, phospho-p38, phospho-JNK and phospho-cJun). Do these antibodies interact with non-phosphorylated protein at certain background level?
These antibodies were specially designed to recognize the phosphorylated forms over the non-phosphorylated forms by the providers. Both the websites and the literature were checked carefully before to select these antibodies.
How many times western-blots with samples shown on Figures 6a and Figure S3 were repeated?
“n= 3 independent experiments.” was added to the respective legends. All these blots were provided as supplementary material.
How the authors would explain differences in kinetics at long times (longer than 200 minutes) seen in Figure 7 C (mitoGCAMP2) for db/+ strain with analogous experiment done in Figure 3a (where there is a rise of calcium after 200 min after crush)?
While the release of calcium in the cytoplasm is a constant feature following the triggering of demyelination in mSC, the hypercalciema that we detected after 2hrs and shown in Fig.3a is not always occurring as illustrated in Fig. 7c. We believe this may due to the physiological state of the cells in some animals but we have no idea of the cause.
These sentences were added to the discussion.
Page 17, lines 17, 21 - Figures S4 and S5 are absent in supplementary information.
As we found data related to Figure S4 and S5 weakly relevant, these data were removed from the manuscript. References to these data are therefore deleted in the updated manuscript.
Page 17, line 38 “Figure 7i left panel ” This statement should be clarified.
We modified the paragraph as following:
“Finally, we investigated the mouse motor behavior using Rotarod and grip tests. Animals were treated daily with the drug or vehicle for 4 weeks starting at the age of one month. These animals were tested one week and 4 weeks after treatment started. We found that muscular strength was affected in diabetic mice and a 4 weeks treat-ment with TRO19622 reversed this defect (Figure 7h). Diabetic mice also performed more poorly on the Rotarod than control heterozygous mice and the treatment with TRO19622 significantly improved their performances vs non-treated db/db mice after a 4 weeks treatment (Figure 7i). We then stopped the treatment and tested animals against 8 weeks later. In these conditions, the observed benefit of TRO19622 treatment was lost (Figure 7i left panel).”
The authors should provide a justification that a required amount of animals was used for the experiments from the point of view of statistical analysis applied.
We added this sentence to the Material and Methods section Data and Statistical analysis: “The size of the animal groups was determined using BioStatTGV (https://biostatgv.sentiweb.fr).”
Reviewer 2 Report
Well written paper showing crush peripheral nerve injury and diabetic Schwann cell demyelination is triggered by Ca release through VD anion Chanel 1
Minor edit change
Page 15 line 4 Two months old mice change to two month old
Author Response
Page 15 line 4 Two months old mice change to two month old
The sentence was modified as following : “Two-month-old mice were treated with 3mg/kg of TRO19622 or vehicle subcutaneously for 5 consecutive days.”
Reviewer 3 Report
This MS is really interesting,
I would suggest to write a clear hypothesis in the Intro section
"Looking at the mitochondrial length, we classified 30 them as short (0.1 to 1m), medium (1 to 3 m) and long (>3 m)(Figure 2b). " How did you choose these criteria?
Author Response
I would suggest to write a clear hypothesis in the Intro section
We added the following sentence to the introduction: “We hypothesized that mitochondria participate to this signaling.”
"Looking at the mitochondrial length, we classified 30 them as short (0.1 to 1mm), medium (1 to 3 mm) and long (>3 mm)(Figure 2b). " How did you choose these criteria?
This sentence has been completed as following : “Looking at the mitochondrial length, we classified them as short (0.1 to 1mm), medium (1 to 3 mm) and long (>3 mm)(Figure 2b) accordingly to their respective average speed in mSC [27].”
This manuscript is a resubmission of an earlier submission. The following is a list of the peer review reports and author responses from that submission.